# ENSURING DNN SOLUTION FEASIBILITY FOR OPTIMIZATION PROBLEMS WITH LINEAR CONSTRAINTS

**Tianyu Zhao[1,2], Xiang Pan[2], Minghua Chen[3,*], Steven H. Low[4]**
[1]Lenovo Machine Intelligence Center, [2]The Chinese University of Hong Kong,
[3]City University of Hong Kong, [4]California Institute of Technology
`tzhao3@lenovo.com`, `Xpan@link.cuhk.edu.hk`,
`minghua.chen@cityu.edu.hk` (*Corresponding author), `slow@caltech.edu`

## ABSTRACT

We propose *preventive learning* as the first framework to guarantee Deep Neural Network (DNN) solution feasibility for optimization problems with linear constraints without post-processing, upon satisfying a mild condition on constraint calibration. Without loss of generality, we focus on problems with only inequality constraints. We systematically calibrate the inequality constraints used in training, thereby anticipating DNN prediction errors and ensuring the obtained solutions remain feasible. We characterize the calibration rate and a critical DNN size, based on which we can directly construct a DNN with provable solution feasibility guarantee. We further propose an *Adversarial-Sample Aware* training algorithm to improve its optimality performance. We apply the framework to develop DeepOPF+ for solving essential DC optimal power flow problems in grid operation. Simulation results over IEEE test cases show that it outperforms existing strong DNN baselines in ensuring 100% feasibility and attaining consistent optimality loss (<0.19%) and speedup (up to ×228) in both light-load and heavy-load regimes, as compared to a state-of-the-art solver. We also apply our framework to a non-convex problem and show its performance advantage over existing schemes.

## 1 INTRODUCTION

Recently, there have been increasing interests in employing neural networks, including deep neural networks (DNN), to solve constrained optimization problems in various problem domains, especially those needed to be solved repeatedly in real-time. The idea behind these machine learning approaches is to leverage the universal approximation capability of DNNs (Hornik et al., 1989; Leshno et al., 1993) to learn the mapping between the input parameters to the solution of an optimization problem. Then one can directly pass the input parameters through the trained DNN to obtain a quality solution much faster than iterative solvers. For example, researchers have developed DNN schemes to solve essential optimal power flow problems in grid operation with sub-percentage optimality loss and several orders of magnitude speedup as compared to conventional solvers (Pan et al., 2020a;b; Donti et al., 2021; Chatzos et al., 2020; Lei et al., 2020). Similarly, DNN schemes also obtain desirable results for real-time power control and beam-forming design (Sun et al., 2018; Xia et al., 2019) problems in wireless communication in a fraction of time used by existing solvers.

Despite these promising results, however, a major criticism of DNN and machine learning schemes is that they usually cannot guarantee the solution feasibility with respect to all the inequality and equality constraints of the optimization problem (Zhao et al., 2020; Pan et al., 2020b). This is due to the inherent neural network prediction errors. Existing works address the feasibility concern mainly by incorporating the constraints violation (e.g., a Lagrangian relaxation to compute constraint violation with Lagrangian multipliers) into the loss function to guide THE DNN training. These endeavors, while generating great insights to the DNN design and working to some extent in case studies, can not guarantee the solution feasibility without resorting to expensive post-processing procedures, e.g., feeding the DNN solution as a warm start point into an iterative solver to obtain a feasible solution. See Sec. 2 for more discussions. To date, it remains a largely open issue of ensuring DNN solution (output of DNN) feasibility for constrained optimization problems.

In this paper, we address this challenge for general Optimization Problems with Linear (inequality) Constraints (OPLC) with varying problem inputs and fixed objective/constraints parameters. Since linear equality constraints can be exploited to reduce the number of decision variables without losing optimality (and removed), it suffices to focus on problems with inequality constraints. Our idea is to train DNN in a preventive manner to ensure the resulting solutions remain feasible even with prediction errors, thus avoiding the need of post-processing. We make the following contributions:

▷ After formulating OPLC in Sec. 3, we propose *preventive learning* as the first framework to ensure the DNN solution feasibility for OPLC without post-processing in Sec. 4. We systematically calibrate inequality constraints used in DNN training, thereby anticipating prediction errors and ensuring the resulting DNN solutions (outputs of the DNN) remain feasible.

▷ We characterize the calibration rate allowed in Sec. 4.1, i.e., the rate of adjusting (reducing) constraints limits that represents the room for (prediction) errors without violating constraints, and a sufficient DNN size for ensuring DNN solution feasibility in Sec. 4.2. We then directly construct a DNN with provably guaranteed solution feasibility.

▷ Observing the feasibility-guaranteed DNN may not achieve strong optimality result, in Sec. 4.3, we propose an adversarial training algorithm, called *Adversarial-Sample Aware* algorithm to further improve its optimality without sacrificing feasibility guarantee and derive its performance guarantee.

▷ We apply the framework to design a DNN scheme, DeepOPF+, to solve DC optimal power flow (DC-OPF) problems in grid operation. Simulation results over IEEE 30/118/300-bus test cases show that it outperforms existing strong DNN baselines in ensuring 100% feasibility and attaining consistent optimality loss ($<0.19\%$) and speedup (up to $\times228$) in both light-load and heavy-load regimes, as compared to a state-of-the-art solver. We also apply our framework to a non-convex problem and show its performance advantage over existing schemes.

## 2 RELATED WORK

There have been active studies in employing machine learning models, including DNNs, to solve constrained optimizations directly (Kotary et al., 2021b; Pan et al., 2019; 2020b; Zhou et al., 2022; Guha et al., 2019; Zamzam & Baker, 2020; Fioretto et al., 2020; Dobbe et al., 2019; Sanseverino et al., 2016; Elmachtoub & Grigas, 2022; Huang et al., 2021; Huang & Chen, 2021), obtaining close-to-optimal solution much faster than conventional iterative solvers. However, these schemes usually cannot guarantee solution feasibility w.r.t. constraints due to inherent prediction errors.

Some existing works tackle the feasibility concern by incorporating the constraints violation in DNN training (Pan et al., 2020a; Donti et al., 2021). In (Nellikkath & Chatzivasileiadis, 2021; 2022), physics-informed neural networks are applied to predict solutions while incorporating the KKT conditions of optimizations during training. These approaches, while attaining insightful performance in case studies, do not provide solution feasibility guarantee and may resort to expensive projection procedure (Pan et al., 2020b) or post-processing equivalent projection layers (Amos & Kolter, 2017; Agrawal et al., 2019) to recover feasibility. A gradient-based violation correction is proposed in (Donti et al., 2021). Though a feasible solution can be recovered for linear constraints, it can be computationally inefficient and may not converge for general optimizations. A DNN scheme applying gauge function that maps a point in an $l_1$-norm unit ball to the (sub)-optimal solution is proposed in (Li et al., 2022). However, its feasibility enforcement is achieved from a computationally expensive interior-point finder program. There is also a line of work (Ferrari, 2009; ul Abdeen et al., 2022; Qin et al., 2019; Limanond & Si, 1998) focusing on verifying whether the output of a given DNN satisfies a set of requirements/constraints. However, these approaches are only used for evaluation and not capable of obtaining a DNN with feasibility-guarantee and strong optimality. To our best knowledge, this work is the *first* to guarantee DNN solution feasibility without post-processing.

Some techniques used in our study (for constrained problems) are related to those for verifying DNN accuracy against input perturbations for unconstrained classification (Sheikholeslami et al., 2020). Our work also significantly differs from (Zhao et al., 2020) in we can provably guarantee DNN solution feasibility for OPLC and develop a new learning algorithm to improve solution optimality.

## 3 OPTIMIZATION PROBLEMS WITH LINEAR CONSTRAINTS (OPLC)

We focus on the standard OPLC formulated as (Faísca et al., 2007):

$$\min\ f(\boldsymbol{x}, \boldsymbol{\theta}) \quad \text{s.t.} \quad g_j(\boldsymbol{x}, \boldsymbol{\theta}) \triangleq \boldsymbol{a_j^T x} + \boldsymbol{b_j^T \theta} \le e_j,\ j \in \mathcal{E}, \tag{1}$$

$$\text{var.} \quad \underline{x}_k \le x_k \le \bar{x}_k,\ k = 1, \dots, N. \tag{2}$$

$\boldsymbol{x} \in \mathcal{R}^N$ are the decision variables, $\mathcal{E}$ is the set of inequality constraints, and $\boldsymbol{\theta} \in \mathcal{D}$ are the OPLC inputs. Convex polytope $\mathcal{D} = \{\boldsymbol{\theta} \in \mathcal{R}^M | \mathbf{A_\theta}\boldsymbol{\theta} \le \boldsymbol{b_\theta}, \exists \mathbf{x} : (1), (2)\text{ hold}\}$ is specified by matrix $\mathbf{A_\theta}$ and vector $\boldsymbol{b_\theta}$ such that $\forall \boldsymbol{\theta} \in \mathcal{D}$, OPLC in (1)-(2) admits a unique optimum.[1] The OPLC objective $f$ is a general convex/non-convex function. For ease of presentation, we use $g_j(\boldsymbol{x}, \boldsymbol{\theta})$ to denote

---

[1] Our approach is also applicable to non-unique solution and unbounded $\boldsymbol{x}$. See Appendix A for a discussion.

Figure 1: Overview of the preventive learning framework to solve OPLC. The calibration rate is first obtained. The sufficient DNN size in ensuring universal feasibility is then determined, and a DNN model can be constructed directly with universal feasibility guarantee in this step. With the determined calibration rate and sufficient DNN size, a DNN model with enhanced optimality without sacrificing feasibility is obtained using the *Adversarial-Sample Aware* algorithm.

$\boldsymbol{a}_j^T \boldsymbol{x} + \boldsymbol{b}_j^T \boldsymbol{\theta}$.[2] We assume that each $g_j(\boldsymbol{x}, \boldsymbol{\theta}) \leq e_j$ is active for at least one combination of $\boldsymbol{\theta} \in \mathcal{D}$ and $\boldsymbol{x}$ satisfying (2) without loss of generality (otherwise $g_j$ is unnecessary and can be removed). We note that linear equality constraints can be exploited (and removed) to reduce the number of decision variables without losing optimality as discussed in Appendix B, it suffices to focus on OPLC with inequality constraints as formulated in (1)-(2).

The OPLC in (1)-(2) has wide applications in various engineering domains, e.g., DC-OPF problems in power systems (Pan et al., 2019) and model-predictive control problems in control systems (Bemporad et al., 2000). While many numerical solvers, e.g., those based on interior-point methods (Ye & Tse, 1989), can be applied to obtain its solution, the time complexity can be significant and limits their practical applications especially considering the problem input uncertainty under various scenarios. The observation that opens the door for DNN scheme development lies in that solving OPLC is equivalent to learning the mapping from input $\boldsymbol{\theta}$ to optimal solution $\boldsymbol{x}^*(\boldsymbol{\theta})$ (Pan et al., 2020a; Bemporad & Filippi, 2006). Thus, one can leverage the universal approximation capability of DNNs (Hornik et al., 1989; Leshno et al., 1993) to learn the input-solution mapping and apply the trained DNN to obtain the optimal solution for any $\boldsymbol{\theta} \in \mathcal{D}$ with significantly lower time complexity. See a concrete example in (Pan et al., 2020b). While DNN schemes achieve promising speedup and optimality performance, a fundamental challenge lies in ensuring solution feasibility, which is nontrivial due to inherent DNN prediction errors. In the following, we propose preventive learning as the first framework to tackle this issue and design DNN schemes for solving OPLC in (1)-(2).

## 4 PREVENTIVE LEARNING FOR SOLVING OPLC

We propose a preventive learning framework to develop DNN schemes for solving OPLC in (1)-(2). We calibrate inequality constraints used in DNN training, thereby anticipating prediction errors and ensuring the resulting DNN solutions remain feasible. See Fig. 2 for illustrations.

First, in Sec. 4.1, we determine the maximum calibration rate for inequality constraints, so that solutions from a preventively-trained DNN using the calibrated constraints respect the original constraints for all possible inputs. Here we refer the output of the DNN as the DNN solution.

Second, in Sec. 4.2, we determine a sufficient DNN size so that with preventive learning there exists a DNN whose worst-case violation on calibrated constraints is smaller than the maximum calibration rate, thus ensuring DNN solution feasibility, i.e., DNN's output always satisfies (1)-(2) for any input. We construct a provable feasibility-guaranteed DNN model, namely DNN-FG, as shown in Fig. 1.

Third, observing DNN-FG may not achieve strong optimality performance, in Sec. 4.3, we propose an adversarial *Adversarial-Sample Aware* training algorithm to further improve DNN's optimality without sacrificing feasibility guarantee, obtaining an optimality-enhanced DNN as shown in Fig. 1.

### 4.1 INEQUALITY CONSTRAINT CALIBRATION RATE

We calibrate each inequality limit $g_j(\boldsymbol{x}, \boldsymbol{\theta}) \leq e_j, j \in \mathcal{E}$ by a calibration rate $\eta_j \geq 0$:[3]

$$g_j(\boldsymbol{x}, \boldsymbol{\theta}) \leq \hat{e}_j = \begin{cases} e_j (1 - \eta_j), & \text{if } e_j \geq 0; \\ e_j (1 + \eta_j), & \text{otherwise.} \end{cases} \tag{3}$$

---

[2]Multiple scalars $\boldsymbol{b}_j^T \boldsymbol{\theta}, j \in \mathcal{E}$ are correlated via $\boldsymbol{\theta}$. Studying varying $\boldsymbol{a}_j, \boldsymbol{b}_j, e_j$ is also a promising future work.
[3]For $g_j$ with $e_j = 0$, one can add an auxiliary constant $\tilde{e}_j \neq 0$ such that $g_j(\boldsymbol{x}, \boldsymbol{\theta}) + \tilde{e}_j \leq \tilde{e}_j$ for the design and formulation consistency. The choice of $\tilde{e}_j$ can be problem dependent. For example, in our simulation, $\tilde{e}_j$ is set as the maximum slack bus generation for its lower bound limit in OPF discussed in Appendix L.

However, an inappropriate calibration rate could lead to poor performance of DNN. If one adjusts the limits too much, some input $\boldsymbol{\theta} \in \mathcal{D}$ will become infeasible under the calibrated constraints and hence lead to poor generalization of the preventively-trained DNN. To this end, we solve the following bi-level optimization to obtain the maximum calibration rate, such that the calibrated feasibility set of $\boldsymbol{x}$ can still support the input region, i.e., the OPLC in (1)-(2) with a reduced feasible set has a solution for any $\boldsymbol{\theta} \in \mathcal{D}$.

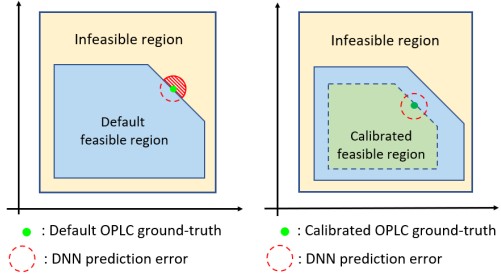

Figure 2: Left: solution of DNN trained with default OPLC ground-truth can be infeasible due to inevitable prediction errors. Right: solution of DNN trained with calibrated OPLC ground-truth ensures universal feasibility even with prediction errors.

$$\min_{\boldsymbol{\theta} \in \mathcal{D}} \max_{\boldsymbol{x}, \nu^c} \quad \nu^c \tag{4}$$

$$\text{s.t. } (1), (2)$$

$$\nu^c \le (e_j - g_j(\boldsymbol{\theta}, \boldsymbol{x}))/|e_j|, \ \forall j \in \mathcal{E}. \tag{5}$$

(1)-(2) enforce the feasibility of $\boldsymbol{x}$ for input $\boldsymbol{\theta} \in \mathcal{D}$. (5) represents the maximum element-wise least redundancy among all constraints, i.e., the maximum constraint calibration rate. Therefore, solving (4)-(5) gives the maximum allowed calibration rate for all inequality constraints and $\boldsymbol{\theta} \in \mathcal{D}$.

It is challenging to solve the above bi-level problem exactly. We apply the following procedure to obtain a lower bound of its optimal objective in polynomial time. See Appendix C for details.

> Step 1. Reformulate the bi-level problem (4)-(5) to an equivalent single-level one by replacing the inner problem with its KKT conditions (Boyd & Vandenberghe, 2004).
>
> Step 2. Transform the single-level optimization problem into a MILP by replacing the bi-linear equality constraints (comes from the complementary slackness in KKT conditions) with equivalent mixed-integer linear inequality constraints.
>
> Step 3. Solve the MILP using the branch-and-bound algorithm (Lawler & Wood, 1966). Let the obtained objective value be $\Delta \ge 0$ from the primal constraint (1) and constraint (5).

**Remark:** (i) the branch-and-bound algorithm returns $\Delta$ (lower bound of the maximum calibration rate $\nu^{c*}$) with a polynomial time complexity of $\mathcal{O}((M + 4|\mathcal{E}| + 5N)^{2.5})$ (Vaidya, 1989), where $M$ and $N$ are the dimensions of the input and decision variables, and $|\mathcal{E}|$ is the number of constraints. (ii) $\Delta$ is a lower bound to the maximum calibration rate as the algorithm may not solve the MILP exactly (by relaxing (some of) the integer variables). Such a lower bound still guarantees that the input region is supported. (iii) If $\Delta = 0$, reducing the feasibility set may lead to no feasible solution for some inputs. (iv) If $\Delta > 0$, we can obtain a DNN with provably solution feasibility guarantee as shown in Sec. 4.2. (v) After solving (4)-(5), we set each $\eta_j$ in (3) to be $\Delta$, such uniform constraints calibration forms the *outer bound* of the minimum supporting calibration region. See Appendix D for a discussion; (vi) we observe that the branch-and-bound algorithm can actually return the exact optimal $\nu^{c*}$ in less than 20 mins for numerical examples studied in Sec. 6.

### 4.2 Sufficient DNN Size for Ensuring Feasibility

In this subsection, we first model DNN with ReLU activations. Then, we introduce a method to determine the sufficient DNN size for guaranteeing solution feasibility.

#### 4.2.1 DNN Model Representation.

We employ a DNN with $N_{\text{hid}}$ hidden layers (depth) and $N_{\text{neu}}$ neurons in each hidden layer (width), using multi-layer feed-forward neural network structure with ReLU activation function to approximate the input-solution mapping for OPLC, which is defined as:

$$\boldsymbol{h_0} = \boldsymbol{\theta}, \ \boldsymbol{h_i} = \sigma\left(\boldsymbol{W_i h_{i-1}} + \boldsymbol{b_i}\right), \ i = 1, \dots, N_{\text{hid}},$$

$$\tilde{\boldsymbol{h}} = \sigma\left(\boldsymbol{W_o h_{N_{\text{hid}}}} + \boldsymbol{b_o} - \underline{\boldsymbol{x}}\right) + \underline{\boldsymbol{x}}, \hat{\boldsymbol{x}} = -\sigma\left(\bar{\boldsymbol{x}} - \tilde{\boldsymbol{h}}\right) + \bar{\boldsymbol{x}}. \tag{6}$$

$\boldsymbol{\theta}$ is the DNN input and $\boldsymbol{h_i}$ is the output of the $i$-th layer. $\boldsymbol{W_i}$ and $\boldsymbol{b_i}$ are the $i$-th layer's weight matrix and bias. $\sigma(x) = \max(x, 0)$ is the ReLU activation, taking element-wise max operation over the input vector. $\tilde{\boldsymbol{h}}$ enforces output feasibility w.r.t. the lower bounds in (2) while final output $\hat{\boldsymbol{x}}$ further satisfies upper bounds. Here we present the last two *clamp*-equivalent actions as (6) for further DNN

analysis. To better include the DNN equations in our designed optimization to analysis DNN's worst case feasibility guarantee performance, we adopt the technique in (Tjeng et al., 2018) to reformulate the ReLU activations expression in (6). For $i = 1, \ldots, N_{\text{hid}}$, let $\hat{h}_i$ denotes $W_i h_{i-1} + b_i$. The output of neuron with ReLU activation is represented as: for $k = 1, \ldots, N_{\text{neu}}$ and $i = 1, \ldots, N_{\text{hid}}$,

$$\hat{h}_i^k \leq h_i^k \leq \hat{h}_i^k - h_i^{\min,k}(1 - z_i^k), \tag{7}$$

$$0 \leq h_i^k \leq h_i^{\max,k} z_i^k, \ z_i^k \in \{0, 1\}. \tag{8}$$

Here we use superscript $k$ to denote the $k$-th element of a vector. $z_i^k$ are (auxiliary) binary variables indicating the state of the corresponding neuron, i.e., 1 (resp. 0) indicates activated (resp. non-activated). We remark that given the value of DNN weights and bias, $z_i^k$ can be determined ($z_i^k$ can be either 0/1 if $\hat{h}_i^k = 0$) for each input $\boldsymbol{\theta}$. $h_i^{\max,k}/h_i^{\min,k}$ are the upper/lower bound on the neuron outputs. See Appendix E.1 for a discussion. With (7)-(8), the input-output relationship in DNN can be represented with a set of mixed-integer linear inequalities. We discuss how to employ (7)-(8) to determine the sufficient DNN size in guaranteeing universal feasibility in Sec. 4.2.2. For ease of representation, we use $(\mathbf{W}, \mathbf{b})$ to denote DNN weights and bias in the following.

### 4.2.2 SUFFICIENT DNN SIZE IN GUARANTEEING UNIVERSAL FEASIBILITY.

As a methodological contribution, we propose an iterative approach to determine the sufficient DNN size for guaranteeing universal solution feasibility in the input region. The idea is to iteratively verify whether the worst-case prediction error of the given DNN is within the room of error (maximum calibration rate), and doubles the DNN's width (with fixed depth) if not. We outline the design of the proposed approach below, under the setting where all hidden layers share the same width. Let the depth and (initial) width of the DNN model be $N_{\text{hid}}$ and $N_{\text{neu}}$, respectively. Here we define *universal solution feasibility* as that for any input $\boldsymbol{\theta} \in \mathcal{D}$, the output of DNN always satisfies (1)-(2).

For each iteration, the proposed approach first evaluates the least maximum relative violations among all constraints for all $\boldsymbol{\theta} \in \mathcal{D}$ for the current DNN model via solving the following bi-level program:

$$\min_{\mathbf{W}, \mathbf{b}} \max_{\boldsymbol{\theta} \in \mathcal{D}} \ \nu^f, \ \text{s.t.} \ (7) - (8), 1 \leq i \leq N_{\text{hid}}, 1 \leq k \leq N_{\text{neu}}, \tag{9}$$

$$\nu^f = \max_{j \in \mathcal{E}}\{(g_j(\boldsymbol{\theta}, \hat{\boldsymbol{x}}) - \hat{e}_j)/|e_j|\}, \tag{10}$$

where (9)-(10) express the outcome of the DNN as a function of input $\boldsymbol{\theta}$. $\nu^f$ is maximum constraints violation degree among all constraints. Thus, solving (9)-(10) gives the least maximum DNN constraint violation over the input region $\mathcal{D}$. We apply gradient descent to solve the above bi-level optimization problem, see Appendix E for details. Let $\rho$ be the obtained objective value of (9)-(10) and $(\mathbf{W}^f, \mathbf{b}^f)$ be the corresponding DNN parameters, with which we can directly construct a DNN model. Recall that the determined calibration rate is $\Delta$, the proposed approach then verifies whether the constructed DNN is sufficient for guaranteeing feasibility by the following proposition.

**Proposition 1** *Consider the DNN with $N_{hid}$ hidden layers each having $N_{neu}$ neurons and parameters $(\mathbf{W}^f, \mathbf{b}^f)$. If $\rho \leq \Delta$, then $\forall \boldsymbol{\theta} \in \mathcal{D}$, the solution generated by this DNN is feasible w.r.t (1)-(2).*

The proof is shown in Appendix F. Proposition 1 states that if $\rho \leq \Delta$, the worst-case prediction error of current DNN model is within the maximum calibration rate and hence the current DNN size is sufficient for guaranteeing universal feasibility; otherwise, it doubles the width of DNN and moves to the next iteration. Recall that the input-solution mapping for OPLC is continuous. Hence, there exists a DNN such that universal feasibility of DNN solution is guaranteed given the DNN size (width) is sufficiently large according to the universal approximation theorem (Hornik, 1991). See Appendix G for the discussion.[4] Details of the procedures are shown in Algorithm 1. After the initialization of DNN model (line 3-4), the proposed approach repeatedly compare the obtained maximum constraints violation ($\rho$) with the calibration rate ($\Delta$), doubles the DNN width (line 5-7), and return the width as $N_{\text{neu}}^*$ until $\rho \leq \Delta$. Thus, we can construct a provable feasibility-guaranteed DNN model by the proposed approach, namely DNN-FG as shown in Fig. 1.

We remark that it is challenging to solve the bi-level problem (9)-(10) exactly, i.e., the obtained $\rho$ is an upper bound of the optimal objective of (9)-(10) in each iteration. Nevertheless, as discussed in the following proposition, the upper bound is still useful for analyzing universal solution feasibility.

---

[4]One can also increase the DNN depth to achieve universal approximation for more degree of freedom in DNN parameters. In this work, we focus on increasing the DNN width for sufficient DNN learning ability.

---

**Algorithm 1:** Determining Sufficient DNN Size

---
1: **Input:** $\Delta$; Initial width $N_{\text{neu}}^{\text{init}}$
2: **Output:** Determined DNN width: $N_{\text{neu}}^*$
3: Set $t = 0$; Set $N_{\text{neu}}^t = N_{\text{neu}}^{\text{init}}$; Obtain $\rho$ via solving (9)-(10)
4: **while** $\rho \geq \Delta$ **do**
5:     Set $N_{\text{neu}}^{t+1} = 2 \times N_{\text{neu}}^t$; Set $t = t + 1$; Solve (9)-(10) and update $\rho$
6: **end while**
7: Set $N_{\text{neu}}^* = N_{\text{neu}}^t$
8: **Return:** $N_{\text{neu}}^*$

---

**Proposition 2** *Assume $\Delta > 0$, Algorithm 1 is guaranteed to terminate in a finite number of iterations. At each iteration $t$, consider the DNN with $N_{hid}$ hidden layers each having $N_{neu}^t$ neurons, we can obtain $\rho$ as an upper bound to the optimal objective of (9)-(10) with a time complexity $\mathcal{O}((M + |\mathcal{E}| + 2N_{hid}N_{neu}^t + 4N)^{2.5})$. If $\rho \leq \Delta$, then the DNN with depth $N_{hid}$ and width $N_{neu}^t$ is sufficient in guaranteeing universal feasibility. Furthermore, one can construct a feasibility-guaranteed DNN with the obtained DNN parameters $(\mathbf{W}^f, \mathbf{b}^f)$ such that for any $\theta \in \mathcal{D}$, the solution generated by this DNN is feasible w.r.t. (1)-(2).*

Proposition 2 indicates $\rho$ can be obtained in polynomial time. If $\rho \leq \Delta$, it means the current DNN size is sufficient to preserve universal solution feasibility in the input region; otherwise, it means the current DNN size may not be sufficient for the purpose and it needs to double the DNN width.

We also remark that the obtained sufficient DNN size may not be the minimal sufficient one if the above bi-level optimization problem is not solved exactly. Please refer to Appendix H for detailed discussions. In our case study in Sec. 6, we observe that the evaluated initial DNN size can always guarantee universal feasibility without constraints violation, and we hence conduct further simulations with such determined sufficient DNN sizes.

### 4.3 ADVERSARIAL-SAMPLE AWARE ALGORITHM

While we can directly construct a feasibility-guaranteed DNN (without training) as shown in Proposition 2, it may not achieve strong optimality performance. To this end, we propose an *Adversarial-Sample Aware (ASA)* algorithm to further improve the optimality performance. The algorithm leverages the ideas of adversarial learning (Chakraborty et al., 2018) and active learning (Ren et al., 2021) techniques, which adaptively incorporates adversarial inputs, i.e., the inputs that cause infeasible DNN solutions, for pursuing strong optimality result while preserving universal feasibility guarantee. We outline the algorithm in the following. Denote the initial training set as $\mathcal{T}^0$, containing randomly-generated input and the corresponding ground-truth obtained by solving the calibrated OPLC (with calibration rate $\Delta$). The proposed *ASA* algorithm first pre-trains a DNN model with the sufficient size determined by the approach discussed in Sec. 4.2.2, using the initial training set $\mathcal{T}^0$ and the following loss function $\mathcal{L}$ for each instance as the supervised learning approach:

$$\mathcal{L} = \frac{w_1}{N} \|\hat{\boldsymbol{x}} - \boldsymbol{x}^*\|_2^2 + \frac{w_2}{|\mathcal{E}|} \sum_{j \in \mathcal{E}} \max(g_j(\hat{\boldsymbol{x}}, \boldsymbol{\theta}) - \hat{e}_j, 0). \tag{11}$$

We leverage the penalty-based training idea in (11). The first term is the mean square error between DNN prediction $\hat{\boldsymbol{x}}$ and the ground-truth $\boldsymbol{x}^*$ provided by the solver for each input. The second term is the inequality constraints violation w.r.t calibrated limits $\hat{e}_j$. $w_1$ and $w_2$ are positive weighting factors to balance prediction error and penalty. Hence, training DNN by minimizing (11) can pursue a strong optimality performance as DNN prediction error is also minimized.

However, traditional penalty-based training by only minimizing (11) can not guarantee universal feasibility (Venzke et al., 2020; Pan et al., 2020b). To address this issue, the *ASA* algorithm repeatedly updates the DNN model with adversarial samples, anticipating the post-trained DNNs can eliminate violations around such inputs. Specifically, given current DNN parameters, it finds the worst-case input $\boldsymbol{\theta}^i \in \mathcal{D}$ by solving the inner maximization problem of (9)-(10). Let $\gamma$ be the obtained objective value. Recall that the calibration rate is $\Delta$. If $\gamma \leq \Delta$, the algorithm terminates; otherwise, it incorporates a subset of samples randomly sampled around $\boldsymbol{\theta}^i$ and solves the calibrated OPLC with $\Delta$, and starts a new round of training. Details of the *ASA* algorithm are shown in Appendix I. We highlight the difference between the DNN obtained in Sec. 4.2.2 and that from *ASA* algorithm as follows. The former is directly constructed via solving (9)-(10), which guarantees universal feasibility whilst without considering optimality. In contrast, the latter is expected to enhance optimality while preserving universal feasibility as both optimality and feasibility are considered during training. We further provide theoretical guarantee of it in ensuring universal feasibility in the following.

**Proposition 3** *Consider a DNN model with $N_{hid}$ hidden layers each having $N_{neu}^*$ neurons. For each iteration $i$, assume such a DNN trained with the ASA algorithm can maintain feasibility at the constructed neighborhood $\hat{\mathcal{D}}^j = \{\boldsymbol{\theta} | \boldsymbol{\theta}^j \cdot (1 - a) \leq \boldsymbol{\theta} \leq \boldsymbol{\theta}^j \cdot (1 + a), \boldsymbol{\theta} \in \mathcal{D}\}$ around $\boldsymbol{\theta}^j$ with some small constant $a > 0$ for $\forall j \leq i$. There exists a constant $C$ such that the algorithm is guaranteed to ensure universal feasibility as the number of iterations is larger than $C$.*

The proof idea is shown in Appendix J. Proposition 3 indicates that, with the iterations is large enough, the *ASA* algorithm can ensure universal feasibility by progressively improving the DNN performance around each region around worst-case input. It provides a theoretical understanding of the justifiability of the *ASA* algorithm. In practice, we can terminate the *ASA* algorithm whenever the maximum solution violation is smaller than the inequality calibration rate, which implies universal feasibility guarantee. We note that the feasibility enforcement in the empirical/heuristic algorithm achieves strong theoretical grounding while its performance can be affected by the training method chosen. Nevertheless, as observed in the case study in Appendix M, the *ASA* algorithm terminates in at most 52 iterations with 7% calibration rate, showing its efficiency in practical application.

## 5 PERFORMANCE ANALYSIS OF THE PREVENTIVE LEARNING FRAMEWORK

### 5.1 UNIVERSAL FEASIBILITY GUARANTEE

We provide the following proposition showing that the preventive learning framework generates two DNN models with universal feasibility guarantees.

**Proposition 4** *Let $\Delta$, $\rho$, and $\gamma$ be the determined maximum calibration rate, the obtained objective value of (9)-(10) to determine the sufficient DNN size, and the obtained maximum relative violation of the trained DNN from Adversarial-Sample Aware algorithm following steps in preventive framework, respectively. Assume (i) $\Delta > 0$, (ii) $\rho \leq \Delta$, and (iii) $\gamma \leq \Delta$. The DNN-FG obtained from determining sufficient DNN size can provably guarantee universal feasibility and the DNN from ASA algorithm further improves optimality without sacrificing feasibility guarantee $\forall \boldsymbol{\theta} \in \mathcal{D}$.*

Proposition 4 indicates the DNN model obtained by preventive learning framework is expected to guarantee the universal solution feasibility, which is verified by simulations in Sec. 6.

### 5.2 RUN-TIME COMPLEXITY

We present the complexity of the traditional method in solving the optimization problems with linear constraints. To the best of our knowledge, OPLC in its most general form is NP-hard cannot be solved in polynomial tie unless P=NP. To better deliver the results here, we consider the specific case of OPLC, namely the mp-QP problem, with linear constraints and quadratic objective function. We remark that the complexity of solving mp-QP provides a lower bower for the general OPLC problem. Under this setting, the DNN based framework has a complexity of $\mathcal{O}\left(N^2\right)$ whilst the best known iterative algorithm (Ye & Tse, 1989) requires $\mathcal{O}\left(N^4 + |\mathcal{E}|M\right)$. This means that the computational complexity of the proposed framework is lower than that of traditional algorithms. The comparison results demonstrate the efficiency of the preventive learning framework. See Appendix K for details.

### 5.3 TRADE-OFF BETWEEN FEASIBILITY AND OPTIMALITY

We remark that to guarantee universal feasibility, the preventive learning framework shrinks the feasible region used in preparing training data. Therefore, the DNN solution may incur a larger optimality loss due to the (sub)-optimal training data. It indicates a trade-off between optimality and feasibility, i.e., larger calibration rate leads to better feasibility but worse optimality. To further enhance DNN optimality performance, one can choose a smaller calibration rate than $\Delta$ while enlarging DNN size for better approximation ability and hence achieve satisfactory optimality and guarantee universal feasibility simultaneously.

## 6 APPLICATION IN SOLVING DC-OPF AND NON-CONVEX OPTIMIZATION

### 6.1 DC-OPF PROBLEM AND DEEPOPF+

DC-OPF is a fundamental problem for modern grid operation. It aims to determine the least-cost generator output to meet the load in a power network subject to physical and operational constraints. With the penetration of renewables and flexible load, the system operators need to handle significant uncertainty in load input during daily operation. They need to solve DC-OPF problem under many scenarios more frequently and quickly in a short interval, e.g., 1000 scenarios in 5 minutes, to obtain a stochastically optimized solution for stable and economical operations. However, iterative solvers may fail to solve a large number of DC-OPF problems for large-scale power networks fast enough for the purpose. Although recent DNN-based schemes obtain close-to-optimal solution much faster than conventional methods, they do not guarantee solution feasibility. We design DeepOPF+ by employing the preventive learning framework to tackle this issue. Consider the DC-OPF formulation:

$$\min_{\boldsymbol{P_G}, \Phi} \sum_{i \in \mathcal{G}} c_i \left( P_{Gi} \right) \quad \text{s.t.} \ \boldsymbol{P_G^{\min}} \leq \boldsymbol{P_G} \leq \boldsymbol{P_G^{\max}}, \mathbf{M} \cdot \Phi = \boldsymbol{P_G} - \boldsymbol{P_D}, -\boldsymbol{P_{\text{line}}^{\max}} \leq \mathbf{B}_{\text{line}} \cdot \Phi \leq \boldsymbol{P_{\text{line}}^{\max}}, \quad (12)$$

$\boldsymbol{P_G^{\min}} \in \mathcal{R}^{|\mathcal{B}|}$ (resp. $\boldsymbol{P_G^{\max}}$) and $\boldsymbol{P_{\text{line}}^{\max}} \in \mathcal{R}^{|\mathcal{K}|}$ are the minimum (resp. maximum) generation limits of generators[5] and branch flow limits of the set of transmission lines denoted as $\mathcal{K}$. $\mathcal{G}, \mathcal{B}$, $\mathbf{M}, \mathbf{B}_{\text{line}}$, and $\Phi \in \mathcal{R}^{|\mathcal{B}|}$ denote the set of generators, buses, bus admittance matrix, line admittance matrix, and bus phase angles, respectively. The objective is the total generation cost and $c_i \left( \cdot \right)$ is the cost function of each generator, which is usually strictly quadratic (Park et al., 1993; tpc, 2018) from generator's heat rate curve. Constraints in (12) enforce nodal power balance equations and the limits on active power generation and branch flow. DC-OPF is hence a quadratic programming and admits a unique optimal solution w.r.t. load input $\boldsymbol{P_D}$. Analogy to OPLC (1)-(2), $\sum_{i \in \mathcal{G}} c_i \left( P_{Gi} \right)$ is the objective function $f$ in (1). $\boldsymbol{P_D}$ is the problem input $\theta$ and $(\boldsymbol{P_G}, \Phi)$ are the decision variables $\boldsymbol{x}$.

We apply the proposed preventive-learning framework to design a DNN scheme, named Deep-OPF+, for solving DC-OPF problems. We refer interested readers to Appendix L for details. Denote $\Delta$, $\rho$, and $\gamma$ as the obtained maximum calibration rate, the obtained objective value of (9)-(10) to determine sufficient DNN size, and the maximum relative violation of the trained DNN from *Adversarial-Sample Aware* algorithm in DeepOPF+ design, respectively. We highlight the feasibility guarantee and computational efficiency of DeepOPF+ in following proposition.

**Corollary 1** *Consider the DC-OPF problem and DNN model defined in (6). Assume (i) $\Delta > 0$, (ii) $\rho \leq \Delta$, and (iii) $\gamma \leq \Delta$, then the DeepOPF+ generates a DNN guarantees universal feasibility for any $\boldsymbol{P_D} \in \mathcal{D}$. Suppose the DNN width is the same order of number of bus $B$, then DeepOPF+ has a smaller computational complexity of $\mathcal{O}\left( B^2 \right)$ compared with that of state-of-the-art iterative methods $\mathcal{O}\left( B^4 \right)$, where $B$ is the number of buses.*

Corollary 1 says that DeepOPF+ can solve DC-OPF with universal feasibility guarantee at lower computational complexity compared to conventional iterative solvers,[6] as DNNs with width $\mathcal{O}(B)$ can achieve desirable feasibility/optimality. Such an assumption is validated in existing literature (Pan et al., 2019) and our simulation. To our best knowledge, DeepOPF+ is the first DNN scheme to solve DC-OPF with solution feasibility guarantee without post-processing. We remark that the DeepOPF+ design can be easily generalized to other linearized OPF models (Cain et al., 2012; Yang et al., 2018; Bolognani & Dörfler, 2015) .

## 6.2 Performance Evaluation over IEEE Test Cases

We evaluate its performance over IEEE 30-/118-/300- bus test cases (tpc, 2018) on the input load region of $[100\%, 130\%]$ of the default load covering both the light-load ($[100\%, 115\%]$) and heavy-load ($[115\%, 130\%]$) regimes, respectively. We conduct simulations in CentOS 7.6 with a quad-core (i7-3770@3.40G Hz) CPU and 16GB RAM. We compare DeepOPF+ with five baselines on the same training/test setting: (i) Pypower: the conventional iterative OPF solver; (ii) DNN-P: A DNN scheme adapted from (Pan et al., 2019). It learns the load-solution mapping using penalty approach without constraints calibration and incorporates a projection post processing if the DNN solution is infeasible; (iii) DNN-D: A penalty-based DNN scheme adapted from (Donti et al., 2021). It includes a correction step for infeasible solutions in training/testing; (iv) DNN-W: A hybrid method adapted from (Dong et al., 2020a). It trains a DNN to predict the primal and dual variables as the warm-start points to the conventional solver; (v) DNN-G: A gauge-function based DNN scheme adapted from (Li et al., 2022). It enforces solution feasibility by first solving a linear program to find a feasible interior point, and then constructing the mapping between DNN prediction in an $l_\infty$ unit ball and the optimum. For better evaluation, we implement two DeepOPF+ schemes with different DNN sizes and calibration rate (3%, 7%) that are all within the maximum allowable one, namely DeepOPF+-3, and DeepOPF+-7. The detailed designs/results are presented in Appendix M.

We use the following performance metrics: (i) the percentage of the feasible solution obtained by DNN, (ii) the average relative optimality difference between the objective values obtained by DNN and Pypower, (iii) the average speedup, i.e., the average running-time ratios of Pypower to DNN-

---

[5]$P_{G_i} = P_{G_i}^{\min} = P_{G_i}^{\max} = 0, \forall i \notin \mathcal{G}$, and $P_{D_i} = 0, \forall i \notin \mathcal{A}$, where $\mathcal{A}$ denotes the set of load buses.

[6]We remark that the training of DNN is conducted offline; thus, its complexity is minor as amortized over many DC-OPF instances, e.g., 1000 scenarios per 5 mins. Meanwhile, the extra cost to solve the new-introduced programs in our design is also minor observing that existing solvers like Gurobi can solve the problems efficiently, e.g., <20 minutes to solve th MILPs to determine calibration rate and DNN size. Thus, we consider the run-time complexity of the DNN scheme, which is widely used in existing studies.

Table 1: Performance comparison with SOTA DNN schemes in light-load and heavy-load regimes.

| Case | Scheme | Average speedups | | Feasibility rate (%) | | Optimality loss (%) | | Worst-case violation (%) | |
|------|--------|------------|------------|------------|------------|------------|------------|------------|------------|
| | | light-load | heavy-load | light-load | heavy-load | light-load | heavy-load | light-load | heavy-load |
| Case30 | DNN-P | ×85 | ×86 | 100 | 88.12 | 0.02 | 0.03 | 0 | 5.43 |
| | DNN-D | ×85 | ×84 | 100 | 93.36 | 0.02 | 0.03 | 0 | 11.19 |
| | DNN-W | ×0.90 | ×0.86 | 100 | 100 | 0 | 0 | 0 | 0 |
| | DNN-G | ×24 | ×26 | 100 | 100 | 0.13 | 0.04 | 0 | 0 |
| | DeepOPF+-3 | ×86 | ×92 | 100 | 100 | 0.03 | 0.04 | 0 | 0 |
| | DeepOPF+-7 | ×86 | ×93 | 100 | 100 | 0.03 | 0.09 | 0 | 0 |
| Case118 | DNN-P | ×137 | ×125 | 68.84 | 54.92 | 0.17 | 0.21 | 19.5 | 44.8 |
| | DNN-D | ×138 | ×124 | 73.42 | 55.37 | 0.20 | 0.24 | 16.69 | 43.1 |
| | DNN-W | ×2.08 | ×2.26 | 100 | 100 | 0 | 0 | 0 | 0 |
| | DNN-G | ×26 | ×16 | 100 | 100 | 1.29 | 0.39 | 0 | 0 |
| | DeepOPF+-3 | ×201 | ×226 | 100 | 100 | 0.18 | 0.19 | 0 | 0 |
| | DeepOPF+-7 | ×202 | ×228 | 100 | 100 | 0.37 | 0.41 | 0 | 0 |
| Case300 | DNN-P | ×115 | ×98 | 91.29 | 78.42 | 0.06 | 0.08 | 261.1 | 443.0 |
| | DNN-D | ×115 | ×102 | 91.99 | 82.92 | 0.07 | 0.07 | 231.6 | 348.1 |
| | DNN-W | ×1.04 | ×1.08 | 100 | 100 | 0 | 0 | 0 | 0 |
| | DNN-G | ×2.44 | ×2.65 | 100 | 100 | 0.32 | 0.06 | 0 | 0 |
| | DeepOPF+-3 | ×129 | ×136 | 100 | 100 | 0.03 | 0.03 | 0 | 0 |
| | DeepOPF+-7 | ×130 | ×138 | 100 | 100 | 0.10 | 0.06 | 0 | 0 |

\* Feasibility rate and Worst-case violation are the results *before* post-processing. Feasibility rates (resp Worst-case violation) after post-processing are 100% (resp 0) for all DNN schemes. We hence report the results before post-processing to better show the advantage of our design. Speedup and Optimality loss are the results *after* post-processing of the final obtained feasible solutions.

\* The *correction* step in DNN-D (with $10^{-3}$ rate) takes longer time compared with $l_1$-projection in DNN-P, resulting in lower speedups.

\* We empirically observe that DNN-G requires more training epochs for satisfactory performance. We report its best results at 500 epochs for Case118/300 in heavy-load and the results at 400 epochs for the other cases. The training epochs for the other DNN schemes are 200.

based approach for the test instances, respectively. (iv) the worst-case violation rate, i.e., the largest constraints violation rate of DNN solutions in the entire load domain.

### 6.2.1 PERFORMANCE COMPARISONS BETWEEN DEEPOPF+ AND EXISTING DNN SCHEMES.

The results are shown in Table 1 with the following observations. First, DeepOPF+ improves over DNN-P/DNN-D in that it achieves consistent speedups in both light-load and heavy-load regimes. DNN-P/DNN-D achieves a lower speedup in the heavy-load regime than in the light-load regime as a large percentage of its solutions are infeasible, and it needs to involve a post-processing procedure to recover the feasible solutions. Note that though DNN-P/DNN-D may perform well on the test set in light-load regime with a higher feasibility rate, its worst-case performance over the entire input domain can be significant, e.g., more than $443\%$ constraints violation for Case300 in the heavy-load region. In contrast, DeepOPF+ guarantees solution feasibility in both light-load and heavy-load regimes, eliminating the need for post-processing and hence achieving consistent speedups. Second, though the warm-start/interior point based scheme DNN-W/DNN-G ensures the feasibility of obtained solutions, they suffer low speedups/large optimality loss. As compared, DeepOPF+ achieves noticeably better speedups as avoiding the iterations in conventional solvers. Third, the optimality loss of DeepOPF+ is minor and comparable with these of the existing state-of-the-art DNN schemes, indicating the effectiveness of the proposed *Adversarial-Sample Aware* training algorithm. Fourth, we observe that the optimality loss of DeepOPF+ increases with a larger calibration rate, which is consistent with the trade-off between optimality and calibration rate discussed in Sec. 5.3. We remark that DC-OPF is an approximation to the original non-convex non-linear AC-OPF in power grid operation under several simplifications. DC-OPF is widely used for its convexity and scalability. Expanding the work to AC-OPF is a promising future work as discussed in Appendix B.

Moreover, we apply our framework to a non-convex problem in (Donti et al., 2021) and show its performance advantage over existing schemes. Detailed design/results are shown in Appendix N.

## 7 CONCLUDING REMARKS

We propose preventive learning as the first framework to develop DNN schemes to solve OPLC with solution feasibility guarantee. Given a sufficiently large DNN, we calibrate inequality constraints used in training, thereby anticipating DNN prediction errors and ensuring the obtained solutions remain feasible. We propose an *Adversarial-Sample Aware* training algorithm to improve DNN's optimality. We apply the framework to develop DeepOPF+ to solve and DC-OPF problems in grid operation. Simulations show that it outperforms existing strong DNN baselines in ensuring feasibility and attaining consistent optimality loss and speedup in both light-load and heavy-load regimes. We also apply our framework to a non-convex problem and show its performance advantage over existing schemes. We remark that the proposed scheme can work for large-scale systems because of the desirable scalability of DNN. Future directions include extending the framework to general non-linear constrained optimization problems like ACOPF and evaluating its performance over systems with several thousand buses and realistic loads as discussed in Appendix B.

ACKNOWLEDGEMENTS

The work presented in this paper was supported in part by a General Research Fund from Research Grants Council, Hong Kong (Project No. 11203122), an InnoHK initiative, The Government of the HKSAR, and Laboratory for AI-Powered Financial Technologies. We thank Dr. Andreas Venzke for his contributions to some initial results and the discussions related to the fully-developed results presented in the paper. We would also like to thank the anonymous reviewers and program committee of ICLR 2023 for giving insightful comments on this article.

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
