# OpenReview forum: "Ensuring DNN Solution Feasibility for Optimization Problems with Linear Constraints"
_ICLR.cc/2023/Conference — ICLR 2023 notable top 25%_

### Official Review · Reviewer_crFR · 2022-10-24

**Confidence:** 3
**Correctness:** 2
**Technical Novelty And Significance:** 2
**Empirical Novelty And Significance:** 3
**Recommendation:** 8

**Clarity, Quality, Novelty And Reproducibility:**

* Are unbounded variables permitted in the model (1-2)?
* "We remark that given the value of DNN weights and bias, the value of z^k_i is uniquely determined by the status of each neuron." This is not true at the decision boundary (e.g., the pre-activation value is zero into a ReLU activation).
* In the computational section: What are the preprocessing times required to set up and configure DeepOPF+ (e.g., training time for each DNN architecture tried, MILP solving time to compute delta, solving time to compute the DNN size, etc.). How does it amortize over the whole family of instances used for evaluation?

**Details Of Ethics Concerns:**

The submission has substantial overlap with the following arxiv submission: << REMOVED BY PCs>>

**Strength And Weaknesses:**

The approach of the paper is a reasonable one, and stitches together results from a number of diverse areas in discrete optimization, deep learning, and power systems. However, I have some concerns about a number of technical errors or confusing mathematical informalities, and also feel that the authors oversell the generality of their proposed method.

* The authors assume that (1-2) admits a unique optimum for each value for theta. Is this a reasonable assumption? If so, why? It seems quite restrictive to me (for example, most "interesting" families of linear optimization problems will not satisfy this), and seems very dependent on the choices of script{D} and the objective. Please provide some examples, or some explanation of how you would verify this property a priori, or an explanation of how the algorithm proceeds without this guarantee. Additionally, on p8 the authors state that "In general, DC-OPF is a quadratic programming problem and admits a unique optimal solution w.r.t. load input". Can the authors provide a citation or proof to this effect? What conditions are placed on the objective and script{D} to make this result hold? I am skeptical such a result will hold if the quadratic objective is simply the zero function, for example.

* The authors state that "it suffices to focus on OPLC with inequality constraints". The justification in Appendix B is that any equations can be projected out to deal only with the original variables. This is true, though the authors should take care to consider implied equations. Note that implied can appear conditionally based on the parameter values: consider a simple example with the constraints x <= 2 - theta and x >= theta, with script{D} = {theta in [0, 1]}.

* I am very confused by Remark (i) on p4: the reference points to a paper proving a time complexity bound for linear programming, but the problem under consideration is a _mixed-integer linear programming problem_. In general these will be NP-hard to solve.

* The abstract is promising too much: The "guarantee...[of] solution feasibility for optimization problems with linear constraints" is contingent on the existence and algorithmic discovery of quantities with certain properties that may not exist (and it seems difficult to prove a priori _if_ they will exist). Moreover, the paper focuses on learning over families of related instances with only variation of the constraint right-hand sides (essentially), which is not mentioned until Section 3. These restrictions are all perfectly fine and justifiable, but they should be made front-and-center.

* I am somewhat skeptical that the proposed algorithm will work well for general OPLC problems. There are seemingly a number of places where the approach can break down; for example: a) if there does not exist a delta > 0 (see "implied equations" point above), b) if such a value for delta exists but cannot be computed in a reasonable amount if time, c) if optimizing a single calibration value in (4-5) for all constraints is too conservative, d) if the DNN size grows needed for a feasibility guarantee is impractically large, or e) if the best solutions lie near the feasible region boundary and are "cut off" by the dilation. The author's open the door for such questions by framing their approach as a general purpose one, and so I think the paper would be more defensible if either 1) the authors present a more diverse set of computational experiments, or 2) tighten the focus of the paper towards DC-OPF.

**Summary Of The Paper:**

The paper presents an algorithm for producing feasible solutions for families of linearly constrained optimization problems with varying right-hand sides. The algorithm can provide feasibility guarantees under certain circumstances by 1) dilating each of the linear constraint (shrinking the feasible region), 2) training a neural network that maps a right-hand side to a "good" point, and then 3) bounding the approximation error of the NN w.r.t. the dilated region to provide a feasibility guarantee w.r.t. the original feasible region. Computational results on optimal power flow problems are provided.

**Summary Of The Review:**

While the method proposed by the authors uses some interesting ideas, I have concerns about the rigor, framing, and generality of the contributions claimed by the authors.

---

> ### Comment · Program_Chairs · 2022-11-06
> **-**
>
> arxiv submission does not violate ICLR policy. To protect double blind review process, PCs removed arxiv link from this review.

---

> ### Author Response · Authors · 2022-11-16
> **Response to Reviewer crFR (1/3)**
>
> We sincerely thank you for reviewing our paper and appreciating our idea and approach. In this response, we address the technical concerns, add detailed simulation results, and expand the simulation setting for comprehensiveness. The modified parts in the paper and supplementary material are marked in BLUE.
>
> **Q1: Unique solution assumption**
>
> **A1:** Thanks for raising this concern. We would like to clarify as follows. First, our assumption is reasonable as many OPLCs’ optimum is unique, given their objective functions are strictly convex. It holds for several practical cases, like DC-OPF problems in power systems [R1] and model-predictive control problems in control systems [R2]. In addition, as proved in [R3], if the optimum is unique, the input-solution mapping is continuous while the DNN function is also continuous, which provides a theoretical grounding of applying DNN to learn such a mapping from the Universal Approximation Theorem of DNN for continuous functions in this work and the existing DNN-based schemes in the literature.
>
> We would like to further discuss the case of non-unique optimums, which is an open problem and challenge of existing end-to-end DNN designs. If OPLC admits multiple optimal solutions for the input, there indeed does not exist an injective mapping between input and solution, i.e., multiple input-solution mappings exist. Consider DNN training in this case, if the ground-truth training data are from different input-solution mappings, DNN could present unsatisfactory performance as solutions to closely related inputs may exhibit large differences, and the learning task can become inherently more difficult [R4,R5,R6]. Nevertheless, our approach is still applicable as the first obtained DNN-FG by determining sufficient DNN size can still guarantee universal feasibility. As introduced in Sec. 4.1 and Sec. 4.2, obtaining calibration rate and determining sufficient DNN size are only related to OPLC constraints. These steps only require obtaining one of the continuous feasible mappings but not optimality. Towards the ASA algorithm, it is straightforward to adopt the approaches in [R4,R5,R6] by improving training data quality, applying the unsupervised learning idea, or leaning the high-dimensional (input+initial point) to optimal solution mapping, which we leave for future work. Finally, our new simulations on non-convex optimization (which can have non-unique optimums) in Appendix N find that the ASA algorithm can still work well, showing the robustness of the design.
>
> For the uniqueness of the DC-OPF solution, it is from the strictly convex objective function [R7,R8]. The positive second-order coefficient is from its heat rate curve; hence the objective is commonly modeled as strictly quadratic. Therefore, its optimal solution is always unique for any feasible input in $D$.
>
> We have added the corresponding discussions in the revision and Appendix A. Hope this explanation can help address the concern.
>
> [R1] X. Pan, et al, "DeepOPF: Deep Neural Network for DC Optimal Power Flow", IEEE SmartGridComm 2019.
>
> [R2] A Bemporad, et al, "The explicit solution of model predictive control via multiparametric quadratic programming." American Control Conference, 2000.
>
> [R3] X. Pan, et al, “DeepOPF: A Feasibility-Optimized Deep Neural Network Approach for AC Optimal Power Flow Problems”, IEEE Systems Journal, 2022.
>
> [R4] J Kotary, et al, "Learning hard optimization problems: A data generation perspective." NeurIPS, 2021.
>
> [R5] W. Huang and M. Chen, "DeepOPF-NGT: A Fast Unsupervised Learning Approach for Solving AC-OPF Problems without Ground Truth", ICML Workshop, 2021.
>
> [R6] X. Pan, et al, "DeepOPF-AL Augmented Learning for Solving AC-OPF Problems with Multiple Load-Solution Mappings", arXiv preprint, 2022.
>
> [R7] Power Systems Test Case Archive, 2018
>
> [R8] J. H. Park, et al, "Economic load dispatch for piecewise quadratic cost function using hopfield neural network", IEEE Transactions on Power Systems, 1993.
>
> **Q2: Implied equation issue**
>
> **A2:** We agree that the inequality will become equality (i.e., inequality is binding) under some situations. From the example raised by the reviewer, at $\theta=1$, the two constraints $x\leq2-\theta$ and $x\geq\theta$ can not be calibrated, i.e., any calibration rate $\Delta>0$ will lead the input $\theta=1$ being infeasible. On page 4 of the paper, we have discussed ' If $\Delta=0$, reducing the feasibility set may lead to no feasible solution for some inputs'. Such a condition implies that the system is too binding, e.g., for DC-OPF, some line/generator must always be at the capacity bounds. However, such restrictive conditions seldom happen in practice for the power system safety operation. Under such a scenario, one can consider a smaller input region $D$ such that the input is not so extreme and there could exist an interior for the input region, which helps the framework work. We have added the corresponding discussion in Appendix C in the revision.

---

> ### Author Response · Authors · 2022-11-16
> **Response to Reviewer crFR (2/3)**
>
> **Q3: Complexity of the branch-and-bound algorithm**
>
> **A3:** We would like to clarify that the proposed polynomial time complexity is for the lower bound $\Delta$ of the maximum calibration rate, i.e., the objective of (4)-(5), by relaxing (some of) the integer variables of the reformulated MILP. As discussed on Page 4, `` $\Delta$ is a lower bound to the maximum calibration rate as the algorithm may not solve the MILP exactly. Such a lower bound still guarantees that the input region is supported''. We have emphasized this part in the revised paper to avoid confusion.
>
> **Q4: Abstract and Generality of the method**
>
> **A4:** Thanks for the comment. We have modified the abstract and the introduction section to state our contributions more clearly. In the abstract, we add ``...guarantee DNN solution feasibility for OPLC without post-processing under certain mild conditions’’. We understand the reviewer's comment of 'variation of the constraint right-hand sides' refer to the varying $\theta$. In Section 1, we add discussion, ''...address this challenge for general OPLC with varying problem inputs and fixed problem objective/constraints parameters...'' to make the description more precise.
>
> Besides the above revision, as discussed in the footnote on page 2, it is also interesting to study the varying $a_j, b_j, e_j$.  We believe our approach is still applicable to such a case while may have additional computational challenges as discussed in Appendix B. Nevertheless, it is also of great interest to study problems whose parameters are not varying. For example, in DC-OPF, $a_j, b_j, e_j$ are determined by power network topology, which will not change significantly over a long time scale, e.g., months to years. Hence, it is reasonable and practical to study OPLC with varying inputs only. We have included the corresponding discussions in Appendix B. Thanks.
>
> We would like to provide clarification for each concern in the comment **Conditions that the approach may break**
>
> **Q5: Existence of calibration rate**
>
> **A5** Please refer to the response to **Q2**.
>
> **Q6: Complexity to calculate $\Delta$**
>
> **A6:** We agree that we need to solve a MILP to derive the calibration rate. 1) In our simulation, as also reported on page 4, the concerned MILP can be efficiently solved to exact within $<$20 minutes. 2) Though we may not solve (4)-(5) exactly, we can still obtain a useful lower bound with a polynomial time complexity as discussed in response to Q3. 3) Solving (4)-(5) is conducted offline; its complexity can indeed be seen as minor ($<0.015$ ms) as amortizing over many instances in real-time operation, e.g., 1000 scenarios per 5 mins over a year. We have added more discussions in Appendix M in the revision. Thanks.
>
> **Q7: Too conservative of the calibration rate**
>
> **A7:** We agree with the reviewer that the uniform calibration rate for each constraint can be conservative, i.e., the calibrated region forms the outer bound of the minimum supporting calibration region as discussed in Appendix D. One can further calibrate the constraints that allow more reducing space while still supporting the input region by the approach in Appendix D. In addition, as shown in our simulation and discussed in Sec. 5.3, one can also adopt a larger DNN size under a smaller calibration rate for better approximation ability to achieve satisfactory optimality and feasibility performance. In our simulation, we observe that a calibration rate of 3\% can help maintain universal feasibility while the maximum allowed one can be as large as 21.6\%.
>
> **Q8: Large DNN size is needed to guarantee feasibility**
>
> **A8:** In our simulation, we observe that the DNN with a 128/64/32 structure can achieve universal feasibility for the Case300 problem with 369 variables and 960 constraints, implying that in practice, the adopted DNN size is acceptable. This could benefit from the strong approximation ability of DNN. The DNN and test case sizes and the detailed configuration time are reported in Appendix M.
>
> In addition, we can always get an upper bound of the worst-case violation of the (large) tested DNN size in polynomial time to determine the sufficient (large) DNN size. Such an upper bound is still useful for analyzing universal solution feasibility as discussed in Proposition 2 in Sec. 4.2.
>
> Moreover, we also propose an efficient binary-search-based approach in Appendix H to further find the minimum sufficient DNN size in case the determined DNN in Algorithm 1 is too large.
>
> Finally, we agree that an impractically large DNN would introduce an additional computational effort/challenge, which can be a potential limitation. It is also an interesting direction for solving the constrained program w.r.t. the DNN parameters and determining the sufficient DNN size more efficiently, which is non-trivial and still an open problem in DNN scheme design and we would like to leave it for future work. We have added the corresponding discussion in Appendix M.

---

> ### Author Response · Authors · 2022-11-16
> **Response to Reviewer crFR (3/3)**
>
> **Q9: Best solutions lie near the feasible region boundary**
>
> **A9:** We would like to clarify that if the best solution lies near the boundary, our approach can still work. As discussed in Sec. 5.3, the preventive learning framework shrinks the feasible region used in preparing training data. Though the optimum may be close to the boundary, we can still obtain a feasible and (sub)-optimal solution for further DNN training as long as the calibration rate is within the maximum allowable one.
>
> **Q10: Additional running example**
>
> **A10:** Thanks for your suggestion. We agree that adding another running example is helpful in understanding the framework. We adopt the non-convex optimization example in [R1] and the detailed design is shown in Appendix N.
>
> | Scheme | Optimality loss (%) | Feasibility rate (%) | Speedups | Worst-case violation (%) |
> |:---:|:---:|:---:|:---:|:---:|
> | DNN-P | 0.40 | 39.8 | 85.7 | 68.3 |
> | DNN-D | 0.42 | 39.8 | 117.0 | 41.5 |
> | DNN-W | 0 | 100 | 1.02 | 0 |
> | DNN-G | 1076.0 | 100 | 87.0 | 0 |
> | Pre-DNN-5 | 0.34 | 100 | 144.9 | 0 |
> | Pre-DNN-10 | 0.67 | 100 | 145.3 | 0 |
>
> As reported, our scheme (**Pre-DNN-5** and **Pre-DNN-10**) outperform the other approaches in guaranteeing universal feasibility with higher speedups and minor optimality loss. The applied DNN size and the total prepossessing time to configure the DNN is also reasonable. Detailed running time and cost are reported in Appendix N. Hope this example makes our design clearer. Thanks.
>
> [R1] P. L. Donti, D. Rolnick and J. Z. Kolter, "DC3: a learning method for optimization with hard constraints", in Proceedings of 9th ICLR, virtual conference, May 3 – 7, 2021.
>
> **Q11: Unbounded variable in (1)-(2)**
>
> **A11:** Thanks for raising this question. The formulation permits unbounded variables. There are two approaches to handle the unbounded variables: 1) setting \underline{x}_i or $\bar{x}_i$ to be some arbitrarily small/large numbers. 2) only includes the bounded constraints into (4)-(5) and (6), e.g., for the variables a) without lower bound, the DNN output is $\hat{x}_i=-\sigma(\bar{x}_i-(W_o h+b_o)_i)+\bar{x}_i$; b) without upper bound $\hat{x}_i=\tilde{h}_i$; c) without both upper and lower bounds, $\hat{x}_i=(W_oh+b_o)_i$. We have added more discussions in the revision in Appendix A, and the new simulations are conducted under the unbounded decision variables in Appendix N. Thanks for pointing it out.
>
>
> **Q12: ReLU activation**
>
> **A12:** Thanks for pointing it out. We agree that if the pre-activation value is zero, $z^k_i$ can take either zero or 1 value. We have modified the description to make it clearer.
>
> **Q13: Complexity of the framework**
>
> **A13:** We have reported the detailed prepossessing times required to set up and configure the framework in each step in Appendix M. The DNN size and the studied IEEE test cases are listed in Table 3 and Table 4 in Appendix M. We understand the reviewer is concerned about the scalability of the approach.
>
> First, we would like to state that in our theoretical analysis, our design can always provide the corresponding useful upper/lower bounds in each step of the framework in polynomial time, which can still be utilized for constraints calibration and DNN performance analysis.
>
> Second, in our simulation, we clarify that the additional training effort is indeed acceptable, e.g., $<21$ seconds to calculate the calibration rate, $<6$ hours to determine the sufficient DNN size, and $<6$ hours to apply the proposed Adversary Sample Aware algorithm to further improve the DNN performance for Case118, which is a case with 274 decision variables and 410 constraints.
>
> Third, though our method takes additional training efforts, 1) it is conducted offline, once the DNN is configured, it can be continuously applied to many test instances such that the complexity is amortized, e.g., $<0.5$ ms for DC-OPF problems if the system operator needs to solve DC-OPF per 5 minutes over 1000 scenarios over a year; 2) as illustrated, the obtained DNN outperforms existing DNN based approaches in avoiding any post-processing and resulting in a lower real-time runtime complexity, showing its advantage.
>
> Finally, we would like to leave how to set up the DNNs more efficiently and accelerate the corresponding steps as future work, which is non-trivial and still an open problem in DNN scheme design. We have added the corresponding discussion in Appendix M. Thanks.
>
> Thanks for your suggestions and appreciation of our approach again. Please let us know if any other information needed.

---

> ### Author Response · Authors · 2022-11-17
> **A kind reminder of the response and looking forward to your reply**
>
> Dear reviewer,
>
> Thanks for your time in reviewing the paper and your valuable comments. We hope to have answered your concerns in our individual response with detailed clarifications, discussions, and new simulations.
>
> Since it is approaching the end of the discussion period, we would be happy to answer any further queries you might have before the deadline. Do let us know if you found our response satisfactory or/and wish to take forward the discussion.
>
> Best regards,
>
> Authors

---

> ### Author Response · Authors · 2022-11-29
> **A gentle reminder and looking forward to more discussions**
>
> Dear Reviewer,
>
> We sincerely thank you again for your time and valuable comments!  Since we have reached the middle stage of the AC-Reviewer-Author discussion phase and are towards the end of the overall discussion phase, we are thinking of sending this note since we have not heard back from you yet regarding our response to your concerns.
>
> In Discussion Stage 1, we have carefully considered your initial advice/questions and provided individual responses with the revised submission based on your constructive suggestions. We would like to briefly summarize the revision/response as follows:
>
> $1.$ We present detailed clarifications on the concerns raised by the reviewer and the corresponding discussions w.r.t. each question are included in the revised manuscript to avoid potential confusion.
>
> $2.$  We add additional simulation results on the studied DC-OPF problem for a better understanding of the advantage of our design.
>
> $3.$  We expand the simulation setting to a general non-convex optimization to further comprehensively illustrate the effectiveness of our approach.
>
> In Discussion Stage 2, we are actively available for further clarification and discussion with you if there are any unclear parts or concerns/questions. We would really appreciate your feedback to make sure the responses and revisions have addressed your concerns, or whether there is a leftover concern we can address. Thanks!
>
> Sincerely,
>
> Authors

---

> ### Author Response · Authors · 2022-12-06
> **Sincerely looking forward to more discussions and your feedback is appreciated**
>
> Dear Reviewer,
>
> We are very thankful for your insightful comments and benefit a lot from them. As the second stage discussion stage is drawing to a close (12/12), we would be happy to take forward the discussion and answer any further questions you might have.
>
> In our response, we have addressed the technical concerns, added detailed simulation results, expanded the simulation setting for comprehensiveness, and sent detailed responses to each of your questions. We would like to briefly summarize the revision/response as follows:
>
> $1.$ We present detailed clarifications on the concerns raised by the reviewer and the corresponding discussions w.r.t. each question are included in the revised manuscript to avoid potential confusion.
>
> $2.$  We add additional simulation results on the studied DC-OPF problem for a better understanding of the advantage of our design.
>
> $3.$  We expand the simulation setting to a general non-convex optimization to further comprehensively illustrate the effectiveness of our approach.
>
> In light of your comments, we believe the revision significantly improves the paper and addresses the concerns. We really look forward to hearing your feedback!
>
> Sincerely,
>
> Authors

---

> ### Author Response · Authors · 2022-12-13
> **A friendly reminder on the last day discussion**
>
> Dear Reviewer,
>
> We just want to send a kind reminder as the rebuttal discussion period is ending soon within one day. We appreciate it if we can have an opportunity to engage with you and continue the discussion. Please let us know whether our response has addressed your concerns and if further clarifications are needed.
>
> Best regards,
>
> Authors

---

### Official Review · Reviewer_ek1R · 2022-10-25

**Confidence:** 4
**Correctness:** 4
**Technical Novelty And Significance:** 4
**Empirical Novelty And Significance:** 3
**Recommendation:** 8

**Clarity, Quality, Novelty And Reproducibility:**

The submission is clearly-written, provides new ideas (including tying together many tools/pieces from across the literature), and is high-quality in both its theoretical justification and its experimental comparisons.

**Strength And Weaknesses:**

Strengths:
* The problem studied is one of importance, as providing fast and feasible approximators for optimization problems is widely applicable (including in power grid optimization, which the authors explicitly study).
* The paper proposes a very interesting framework for addressing this problem, which nicely ties together many tools and pieces from across the literature.
* The experimental comparison is thorough and well-done.
* The writing and presentation is generally clear and well-done.

Weaknesses:
* For the ICLR audience, it is important to explain that DC-OPF is an approximation to the true power grid operation problem - otherwise, they may not be aware (and the applicability to DC-OPF only, rather than also AC-OPF, is a key limitation of the work).
* Corollary 1 states: “Suppose the DNN width is the same order of number of bus $B$.” This assumption is not justified in the text.

Note: I have previously reviewed this paper, and believe the authors have addressed all my major concerns from last time. I think this paper is deserving of publication at ICLR.


**Summary Of The Paper:**

This paper proposes a framework for learning provably feasible approximations to optimization problems with linear constraints. This framework entails
* Rewriting the original optimization problem, if it has (linear) equality constraints, in a solely inequality-constrained form, by using variable reduction techniques,
* Constructing a buffer on the inequality constraints of the problem in a way that still ensures feasibility with respect to the input space,
* Sizing a neural network to be large enough to be able to actually achieve feasibility over this input space (and actually solving for the parameters of such a feasible neural network, in order to initialize the network),
* Training the neural network, using both input data and additional adversarial samples generated during neural network training to improve performance on the optimization objective.

The authors demonstrate the efficacy of their approach on DC optimal power flow, a linearly-constrained optimization problem used in power grids to schedule power generators under (approximate) physical constraints.


**Summary Of The Review:**

This paper proposes a framework for learning provably feasible approximations to optimization problems with linear constraints. The submission is clearly-written, provides new ideas (including tying together many tools/pieces from across the literature), and is high-quality in both its theoretical justification and its experimental comparisons.

---

> ### Author Response · Authors · 2022-11-16
> **Response to Reviewer ek1R**
>
> Dear reviewer, thanks for appreciating the idea and approaches in our paper and your (past) review comments. It does help improve the paper quality a lot. We have further clarified your concerns in the paper and in this response. The modified parts in the paper and supplementary material are marked in BLUE. Thanks for your suggestion and time.
>
> **Q1: Discussion on DC-OPF**
>
> **A1:** Thanks for the suggestion. We discuss the limitation and future work in Sec. 6 as "We remark that DC-OPF is an approximation to the original non-convex non-linear AC-OPF in power grid operation under several simplifications. DC-OPF is widely used for its convexity and scalability. Expanding the work to AC-OPF is a promising future work as discussed in Appendix B.'' and in Appendix B " ...to non-linear inequality constraints, e.g., AC-OPF problems with several thousand buses, but with additional computational challenge in solving the related programs…’’. Thanks for the suggestion.
>
> **Q2: Assumption on the DNN width of O(B)**
>
> **A2:** Thanks for pointing it out. The validity of this assumption is supported by the existing literature and our simulations. We have added more clarifications on it in Sec 6.1, stating ``DNNs with width $O(B)$
> can achieve desirable feasibility/optimality. Such an assumption is validated in existing literature [R1,R2] and our simulation.''
>
> [R1] X. Pan, T. Zhao, M. Chen and S. Zhang, "DeepOPF: A Deep Neural Network Approach for Security-Constrained DC Optimal Power Flow", in IEEE Transactions on Power Systems, vol. 36, no. 3, pp. 1725 - 1735, May. 2021.
>
> [R2] P. L. Donti, D. Rolnick and J. Z. Kolter, "DC3: a learning method for optimization with hard constraints", in Proceedings of 9th International Conference on Learning Representations (ICLR), virtual conference, May 3 – 7, 2021.
>
> Thanks for your suggestions and appreciation of our approach again. Please let us know if any other information needed.

---

> > ### Comment · Reviewer_ek1R · 2022-11-28
> > **Response**
> >
> > Thanks to the authors for their revisions!
> >
> > A quick note: I believe the citation here should be only to Pan et al. 2019, not Donti et al. 2021, as the latter studies only ACOPF:
> > > as DNNs with width O(B) can achieve desirable feasibility/optimality. Such an assumption is validated in existing literature (Pan et al., 2019; Donti et al., 2021)

---

> > > ### Author Response · Authors · 2022-11-28
> > > **Response on the reference cited**
> > >
> > > Dear reviewer,
> > >
> > > Thanks for pointing it out. We have revised the draft and only cite (Pan et al. 2019) in the revision.  We will upload the latest paper in the final version.
> > >
> > > Thanks again for your time and suggestion!
> > >
> > > Best,
> > >
> > > Authors

---

> ### Author Response · Authors · 2022-11-17
> **A kind reminder of the response and looking forward to your reply**
>
> Dear reviewer,
>
> Thanks for your time in reviewing the paper and your valuable comments. We hope to have answered your concerns in our individual response with detailed clarifications, discussions, and new simulations.
>
> Since it is approaching the end of the discussion period, we would be happy to answer any further queries you might have before the deadline. Do let us know if you found our response satisfactory or/and wish to take forward the discussion.
>
> Best regards,
>
> Authors

---

### Official Review · Reviewer_Z4Mb · 2022-10-31

**Confidence:** 2
**Correctness:** 3
**Technical Novelty And Significance:** 2
**Empirical Novelty And Significance:** 3
**Recommendation:** 6

**Clarity, Quality, Novelty And Reproducibility:**

Since this paper proposed the first provably effective framework for using DNN in linear constraint optimization problem without post-processing, I think the contribution of this work is meaningful and novel.

**Strength And Weaknesses:**

Strength:
(1) This paper proposed the first DNN-based approach to solve constrianed problem with linear constraints without post-processing, which could be interesitng to the community.
(2) Most of the steps in the algorithm is supported by theoretical guarantees.
(3) The effectiveness of the proposed method is supported by some empirical results.

Weakness:
I didn't see obvious weakness in this paper.

**Summary Of The Paper:**

This paper proposes the first preventive learning based approach to guarantee DNN solution feasible for optimization problem with linear constrains. The key idea is to obtain an conservative satisfication of constriants and adjusting the size of DNN accordlying to guarantee that the redundant feasible region is large enough to offset the approximation error of DNN. The development of the algorithm is heavily based on bi-level optimization which is difficult to solve exactly and the author proposed several approximation approaches to address these issues.

**Summary Of The Review:**

Overall I think this paper proposed a very interesting approach and most of the results look solid to me. However, since I am not the expert in this areas so I will leave the decision to other reviewers.

---

> ### Author Response · Authors · 2022-11-16
> **Response to Reviewer Z4MB**
>
> Dear reviewer, thanks for appreciating the idea and approaches in our paper. We have further addressed your and the other reviewers' concerns in this response and increased the quality of the paper significantly. The modified parts in the paper and supplementary material are marked in BLUE. Please refer to the following response:
>
> **Q1: Complexity of the framework**
>
> **A1:** Thanks for acknowledging our approach. Besides the approximation results and following other reviewers' comments, we have reported the detailed prepossessing times required to set up and configure the framework in each step in Appendix M. The DNN size and the studied IEEE test cases are listed in Tables 3 and 4 in Appendix M. We understand the reviewer is concerned about the scalability of the approach.
>
> First, we would like to state that in our theoretical analysis, our design can always provide the corresponding useful lower/upper bounds in each step of the framework in polynomial time, which can still be utilized for constraints calibration and DNN performance analysis.
>
> Second, in our simulation, we clarify that the additional training effort is indeed acceptable, e.g., $<21$ seconds to calculate the calibration rate, $<6$ hours to determine the sufficient DNN size, and $<6$ hours to apply the proposed Adversary Sample Aware algorithm to further improve the DNN performance for Case118, which is a case with 274 decision variables and 410 constraints.
>
> Third, though our method takes additional training efforts, 1) it is conducted offline, once the DNN is configured, it can be continuously applied to many test instances such that the complexity is amortized, e.g., $<0.5$ ms for DC-OPF problems if the system operator needs to solve DC-OPF per 5 minutes over 1000 scenarios over a year; 2) as illustrated, the obtained DNN outperforms existing DNN based approaches in avoiding any post-processing and resulting in a lower real-time runtime complexity, showing its advantage.
>
> Finally, we would like to leave how to set up the DNNs more efficiently and accelerate the corresponding steps as future work, which is non-trivial and still an open problem in DNN scheme design. We have added the corresponding discussion in Appendix M. Thanks.
>
> Hope this response help to address the potential concerns you may have. We would be very grateful if the recommendation score could be raised. Please let us know if any other information needed.

---

> ### Author Response · Authors · 2022-11-17
> **A kind reminder of the response and looking forward to your reply**
>
> Dear reviewer,
>
> Thanks for your time in reviewing the paper and your valuable comments. We hope to have answered your concerns in our individual response with detailed clarifications, discussions, and new simulations.
>
> Since it is approaching the end of the discussion period, we would be happy to answer any further queries you might have before the deadline. Do let us know if you found our response satisfactory or/and wish to take forward the discussion.
>
> Best regards,
>
> Authors

---

> ### Author Response · Authors · 2022-11-29
> **A gentle reminder and looking forward to more discussions**
>
> Dear Reviewer,
>
> We sincerely thank you again for your time and valuable comments!  Since we have reached the middle stage of the AC-Reviewer-Author discussion phase and are towards the end of the overall discussion phase, we are thinking of sending this note since we have not heard back from you yet regarding our response.
>
> In Discussion Stage 1, we have carefully considered your and the other reviewers' initial advice/questions and provided individual responses with the revised submission. We would like to briefly summarize the revision/response as follows:
>
> $1.$ We present detailed clarifications on the concerns raised by the reviewers and the corresponding discussions w.r.t. each question are included in the revised manuscript to avoid potential confusion.
>
> $2.$  We add additional simulation results on the studied DC-OPF problem for a better understanding of the advantage of our design.
>
> $3.$  We expand the simulation setting to a general non-convex optimization to further comprehensively illustrate the effectiveness of our approach.
>
> In Discussion Stage 2, we are actively available for further clarification and discussion with you if there are any unclear parts or concerns/questions. We would really appreciate your feedback to make sure the responses and revisions have addressed your concerns, or whether there is a leftover concern we can address. Thanks!
>
> Sincerely,
>
> Authors

---

> ### Author Response · Authors · 2022-12-06
> **Sincerely thank you for your positive comment and a kind reminder for your feedback**
>
> Dear Reviewer,
>
> We are very thankful for your positive comments. As the second stage discussion stage is drawing to a close (12/12), we would like to summarize our revisions and send this kind reminder in case you have not accessed our response yet. We would be happy to take forward the discussion for any further questions you might have.
>
> In our previous revision, we addressed the technical concerns, added detailed simulation results, expanded the simulation setting for comprehensiveness, and detailed responses to your and the other reviewers' questions. We would like to briefly summarize the revision/response as follows:
>
> $1.$ We present detailed clarifications on the concerns raised by the reviewers and the corresponding discussions w.r.t. each question are included in the revised manuscript to avoid potential confusion.
>
> $2.$  We add additional simulation results on the studied DC-OPF problem for a better understanding of the advantage of our design.
>
> $3.$  We expand the simulation setting to a general non-convex optimization to further comprehensively illustrate the effectiveness of our approach.
>
> Thanks again for your positive comments and for appreciating the contribution and novelty of our work! Please do let us know if you wish to keep up the discussion.
>
> Sincerely,
>
> Authors

---

> > ### Comment · Reviewer_Z4Mb · 2022-12-14
> > **Thanks for the reply**
> >
> > Thanks author for the detailed explaination. After reading author's response and other reviews, I will keep my score at 6.

---

> > > ### Author Response · Authors · 2022-12-14
> > > **Thanks for the positive feedback**
> > >
> > > Dear Reviewer,
> > >
> > > Thanks a lot for your positive feedback!
> > >
> > > Sincerely,
> > >
> > > Authors

---

> ### Author Response · Authors · 2022-12-13
> **A friendly reminder on the last day discussion**
>
> Dear Reviewer,
>
> We just want to send a kind reminder as the rebuttal discussion period is ending soon within one day. We appreciate it if we can have an opportunity to engage with you and continue the discussion. Please let us know whether our responses are satisfactory and if further clarifications are needed.
>
> Best regards,
>
> Authors

---

### Official Review · Reviewer_f2t2 · 2022-11-01

**Confidence:** 2
**Correctness:** 3
**Technical Novelty And Significance:** 3
**Empirical Novelty And Significance:** 2
**Recommendation:** 6

**Clarity, Quality, Novelty And Reproducibility:**

Clarity: the paper is overall well-written. It might be helpful to add a small running example of solving a small optimization problem with the proposed techniques.

Quality: the quality could be improved if concrete runtime results could be reported.

Originality: the contribution is original as far as I can tell.

**Strength And Weaknesses:**

Strength:

1. The paper tackles an important problem of guaranteeing correctness of DNN-based solution (i.e., does not violate constraints) for the optimization problem, which is a major limitation of end-to-end neural network solutions.

2. The evaluation suggests that the proposed techniques outperform baseline methods either in feasibility rate or optimality loss.

Weakness:
1. The repeated solving of (9)-(10) seems to be very expensive as it involves repeatedly solving MILPs.

2. It is unclear from the paper how challenging the CD-OPT problem actually are. If would be helpful if the authors report the concrete runtime of Pypower on these problems.

3. It is unclear how scalable the proposed approach is (e.g., how fast the size of the network grows as the number of constraints increases, how long each component of the procedure takes). It would be helpful to report these numbers.

Questions:
1. Could the authors clarify why the branch-and-bound procedure for computing the calibration rate has polynomial time complexity? Isn't MILP solving an NP complete problem?

**Summary Of The Paper:**

This paper proposes a workflow for using neural networks to solve optimisation problems while guaranteeing the feasibility of the solution.

**Summary Of The Review:**

I like the direction this paper is taking and the proposed techniques seem promising. However I find the experimental evaluation a little insufficient to evaluate the effectiveness of the approach and I am concerned with the scalability of the approach.

---

> ### Author Response · Authors · 2022-11-16
> **Response to Reviewer f2t2 (1/2)**
>
> We sincerely thank you for reviewing our paper and appreciating our idea and approach. In this response, we address the technical concerns, add detailed simulation results, and expand the simulation setting for better understanding. The modified parts in the paper and supplementary material are marked in BLUE.
>
> **Q1: Complexity of solving (9)-(10)**
>
> **A1:** We understand the concern that the proposed framework may incur high complexity in practice as we need to repeatedly solve (9)-(10) to determine the sufficient DNN size. To this point, we believe the approach is computationally efficient as explained:
>
> 1. First, as discussed in Proposition 2, even though the existing solvers/algorithms can not solve (9)-(10) exactly, we can still get a useful upper bound $\rho$ on its optimal objective in *polynomial time*. Such an upper bound can be utilized to determine whether the considered DNN size is sufficient to maintain universal feasibility.
>
> 2. Second, in our simulation, we observe that the initial tested DNN size (from educated guess based on some preliminary simulation) is sufficient for achieving universal feasibility, i.e., $\rho\leq\Delta$, such that we indeed do not need the iterations in Algorithm 2 for the application in DC-OPF.
>
> 3. Third, we observe that the extra cost to solve the concerned program in our design is acceptable. Specifically, we observe that existing solvers like IPOPT could solve the problems efficiently, e.g., $<$11 minutes, as reported in Appendix M. The detailed prepossessing times required to set up and configure the framework in each step are also presented in Appendix M.
>
> 4. Fourth, as the involved steps to set up the DNN are conducted offline, its complexity is minor ($<0.2$ ms) as amortized over many DC-OPF instances, e.g., 1000 scenarios per 5 mins over a year. Therefore, though additional training efforts are needed to achieve feasibility guarantee, we believe the design is still efficient in implementation with low real-time run-time complexity.
>
> 5. Finally, we would like to leave the open problem of how to determine the sufficient DNN size for the specific learning task more efficiently as future work. We have added corresponding discussions in Appendix M. Thanks.
>
> **Q2: Runtime complexity of DC-OPF**
>
> **A2:** Thanks for the suggestion. We have included the detailed runtime results for both DC-OPF and DNN schemes in Appendix M. As discussed in Sec. 6, due to increasing uncertainty from renewable generation and flexible load, grid operators now need to solve DC-OPF problems under many scenarios in a short interval, e.g., 1000 scenarios in 1 minute. To obtain a stochastically optimized solution, e.g., $\sim$2 minutes for the iterative solvers to solve a large number of DC-OPF problems for Case118, which is ineffective for real-time operation. In contrast, the developed DNN scheme can return the solution with $\times$228 speedups, i.e., less than 0.6 seconds. We have added the results and corresponding discussions in the revised Appendix M.
>
> **Q3: Scalability of the approach**
>
> **A3:** Thanks for the suggestion. We observe that a 3 layers DNN with a 32/16/8 neuron structure can achieve universal feasibility for Case30 with 36 variables and 94 constraints while a 128/64/32 DNN is sufficient for Case300 with 369 variables and 960 constraints, implying the increase of DNN size is acceptable. We have reported the detailed prepossessing times and DNN sizes required to set up the framework in each step and test case in Appendix M. We would like to further clarify as follows:
>
> First, we would like to state that in our theoretical analysis, it can always provide the corresponding useful lower/upper bounds in each step of the framework in polynomial time, which can still be utilized for constraints calibration and DNN performance analysis.
>
> Second, in our simulation, we clarify that the additional training effort is acceptable. For example, for Case118, with 274 decision variables and 410 constraints, it takes $<21$ seconds to obtain the calibration rate, $<6$ hours to determine the sufficient DNN size, and $<6$ hours to apply the proposed Adversary Sample Aware algorithm to further improve the DNN performance.
>
> Third, though our method takes additional training efforts, 1) it is conducted offline, once the DNN is configured, it can be continuously applied to many test instances such that the complexity is amortized, e.g., $<0.5$ ms for DC-OPF problems if the system operator needs to solve DC-OPF per 5 minutes over 1000 scenarios over a year; 2) as illustrated, the obtained DNN outperforms existing DNN based approaches in avoiding any post-processing and resulting in a lower real-time runtime complexity, showing its advantage.
>
> Finally, we would like to leave how to set up the DNN more efficiently and accelerate the corresponding steps as future work, which is non-trivial and still an open problem in DNN scheme design. We have added the corresponding discussion in Appendix M. Thanks.

---

> > ### Comment · Reviewer_f2t2 · 2022-12-01
> > **Thank you for your response**
> >
> > I thank the authors for the explanation and the additional experiments. My main concern on runtime has been resolved and therefore I will raise my score.

---

> > > ### Author Response · Authors · 2022-12-01
> > > **Thanks for the positive feedback**
> > >
> > > Dear Reviewer,
> > >
> > > Thanks a lot for your positive feedback!
> > >
> > > Sincerely,
> > >
> > > Authors

---

> ### Author Response · Authors · 2022-11-16
> **Response to Reviewer f2t2 (2/2)**
>
>  **Q4: Complexity of the branch-and-bound algorithm**
>
> **A4:** We would like to clarify that the proposed polynomial time complexity is for the lower bound $\Delta$ of the maximum calibration rate, i.e., the objective of (4)-(5), by relaxing (some of) the integer variables of the reformulated MILP. As discussed on Page 4, ``$\Delta$ is a lower bound to the maximum calibration rate as the algorithm may not solve the MILP exactly. Such a lower bound still guarantees that the input region is supported''. We have emphasized this part in the revised paper to avoid confusion.
>
>  **Q5: Additional running example**
>
> **A5:** Thanks for the suggestion. We agree that adding another running example helps understand the framework. We adopt the non-convex optimization example in [R1], and the detailed design is shown in Appendix N.
>
> | Scheme | Optimality loss (%) | Feasibility rate (%) | Speedups | Worst-case violation (%) |
> |:---:|:---:|:---:|:---:|:---:|
> | DNN-P | 0.40 | 39.8 | 85.7 | 68.3 |
> | DNN-D | 0.42 | 39.8 | 117.0 | 41.5 |
> | DNN-W | 0 | 100 | 1.02 | 0 |
> | DNN-G | 1076.0 | 100 | 87.0 | 0 |
> | Pre-DNN-5 | 0.34 | 100 | 144.9 | 0 |
> | Pre-DNN-10 | 0.67 | 100 | 145.3 | 0 |
>
>
> As reported, our scheme (**Pre-DNN-5** and **Pre-DNN-10**) outperform the other approaches in guaranteeing universal feasibility with higher speedups and minor optimality loss. The applied DNN size and the total prepossessing time to configure the DNN is also reasonable. Detailed running time and cost are reported in Appendix N. Hope this example makes our design clearer.
>
> [R1] P. L. Donti, D. Rolnick and J. Z. Kolter, "DC3: a learning method for optimization with hard constraints", in Proceedings of 9th International Conference on Learning Representations (ICLR), virtual conference, May 3 – 7, 2021.
>
> We thank the reviewer for appreciating our design. We hope this response/clarification can help address your concerns with further simulation results and examples. Please let us know if any other information needed.

---

> ### Author Response · Authors · 2022-11-17
> **A kind reminder of the response and looking forward to your reply**
>
> Dear reviewer,
>
> Thanks for your time in reviewing the paper and your valuable comments. We hope to have answered your concerns in our individual response with detailed clarifications, discussions, and new simulations.
>
> Since it is approaching the end of the discussion period, we would be happy to answer any further queries you might have before the deadline. Do let us know if you found our response satisfactory or/and wish to take forward the discussion.
>
> Best regards,
>
> Authors

---

> ### Author Response · Authors · 2022-11-29
> **A gentle reminder and looking forward to more discussions**
>
> Dear Reviewer,
>
> We sincerely thank you again for your time and valuable comments!  Since we have reached the middle stage of the AC-Reviewer-Author discussion phase and are towards the end of the overall discussion phase, we are thinking of sending this note since we have not heard back from you yet regarding our response to your concerns.
>
> In Discussion Stage 1, we have carefully considered your initial advice/questions and provided individual responses with the revised submission based on your constructive suggestions. We would like to briefly summarize the revision/response as follows:
>
> $1.$ We present detailed clarifications on the concerns raised by the reviewer and the corresponding discussions w.r.t. each question are included in the revised manuscript to avoid potential confusion.
>
> $2.$  We add additional simulation results on the studied DC-OPF problem for a better understanding of the advantage of our design.
>
> $3.$  We expand the simulation setting to a general non-convex optimization to further comprehensively illustrate the effectiveness of our approach.
>
> In Discussion Stage 2, we are actively available for further clarification and discussion with you if there are any unclear parts or concerns/questions. We would really appreciate your feedback to make sure the responses and revisions have addressed your concerns, or whether there is a leftover concern we can address. Thanks!
>
> Sincerely,
>
> Authors

---

### Decision · Program_Chairs · 2023-01-20

**Decision:**

Accept: notable-top-25%

**Justification For Why Not Higher Score:**

The support from the reviewers is not strong enough for an oral presentation.

**Justification For Why Not Lower Score:**

The paper has interesting ideas and presents a theoretically inspired heuristic solution to an interesting problem. It also received positive evaluation from the reviewers with higher confidence (although one of the 8's should rather be considered as a 7).

**Metareview: Summary, Strengths And Weaknesses:**

The paper presents a method to train deep neural networks to solve linearly constrained optimization problems so that the predicted solution satisfies the constraints. The algorithm design is supported by theoretical results and the performance is demonstrated in some experimental settings. The paper contains some interesting new ideas (including tying together many tools/pieces from across the literature).

On the negative side, the English of the paper is not satisfactory; for the final version, the authors should carefully check the paper and improve the writing in general. Also, the main text should be somewhat shortened, as the negative vertical spaces used hinder readability at certain places.

Some minor suggestions to improve the writing are listed below:
- p. 2: adversarial training Adversary-Sample Aware algorithm -> adversarial training algorithm, called Adversary-Sample Aware algorithm. Also, "adversarial" seems a better choice than "adversary".
- p. 2: increase the spacing below Fig 1 and Table 1; in general, consider rewriting the paper a bit instead of adding to much negative vertical spacing.
- p. 2: an computational expensive -> a computationally expensive
- p. 4, Step 3: What do you mean by "from primal feasibility (1)." here?
- p. 5: What do you mean by "in our design to analysis the DNN performance"?
- Proposition 1, 2 and other places: What do you mean by "the DNN's solution"? Is it the output of the DNN? It would be good to define it.
- Similarly, define formally what you mean by "universal feasibility" of the solutions predicted by the DNN.

**Note From Pc:**

if the above contains the word "oral" or "spotlight" please see: "oral" presentation means -> notable-top-5% and "spotlight" means -> notable-top-25%. As stated in our emails, we are disassociating presentation type from AC recommendations